# The Mediterranean Diet in Paediatric Gastrointestinal Disorders

**DOI:** 10.3390/nu15010079

**Published:** 2022-12-24

**Authors:** Sabrina Cenni, Veronica Sesenna, Giulia Boiardi, Marianna Casertano, Giovanni Di Nardo, Susanna Esposito, Caterina Strisciuglio

**Affiliations:** 1Department of Experimental Medicine, University of Campania “Luigi Vanvitelli”, 80138 Naples, Italy; 2Department of Medicine and Surgery, Pediatric Clinic, University of Parma, 43126 Parma, Italy; 3Department of Woman, Child and General and Specialist Surgery, University of Campania “Luigi Vanvitelli”, 80138 Naples, Italy; 4Department of Neuroscience, Mental Health and Sense Organs (NESMOS), Faculty of Medicine & Psychology, Sant’Andrea Hospital, Sapienza University of Rome, 00185 Rome, Italy

**Keywords:** Mediterranean diet, inflammatory bowel disease, functional gastrointestinal disorders, coeliac disease

## Abstract

The Mediterranean diet is considered one of the healthiest dietary patterns worldwide, thanks to a combination of foods rich mainly in antioxidants and anti-inflammatory nutrients. Many studies have demonstrated a strong relationship between the Mediterranean diet and some chronic gastrointestinal diseases. The aim of this narrative review was to analyse the role of the Mediterranean diet in several gastrointestinal diseases, so as to give a useful overview on its effectiveness in the prevention and management of these disorders.

## 1. Introduction

The Mediterranean diet (MD) is a term coined by Ancel Keys in the 1960s, describing the dietary pattern of the people living in the Mediterranean area, which is structured on a daily consumption of specific food groups included in a standardized food pyramid [1]. The MD is characterized by a high intake of vegetables, fruits, nuts, legumes, low processed cereals, whole grains, and olive oil, as well as a moderate consumption of fish and poultry, and a low intake of sweets, red meat, and dairy products, especially in the form of aged-cheeses [2,3,4]. The proportionally low content of saturated lipids as well as the high content of monounsaturated fatty acids and omega-3 polyunsaturated fatty acids, due to the use of olive oil as the main seasoning, make the MD a dietary pattern with anti-inflammatory and antioxidant properties [2,5,6,7]. Butyrate and other short-chain fatty acids produced from fibres and starches found in fruits and vegetables, which are prevalent in this type of diet, have a key role in downregulating inflammation and promoting innate immunity [3,8,9]. On the other hand, a low-fibre diet can in fact damage the mucous layer, leading to an increased barrier permeability and direct bacterial translocation across the epithelial membrane [3,8,10]. Moreover, the MD is a low-sugar, low-saturated-lipid, and almost additive-free diet, which t avoids dysbiosis and induces a reduction of the levels of plasmatic inflammatory markers, such as tumour necrosis factor-α, interferon-y and high sensitivity C-reactive protein (hs-CRP) [3,8,9]. On the other hand, the Western diet, with its high-fat intake can worsen the damage, especially in the overweight and obese patient. Indeed, adipose tissue has been demonstrated to be an important source of pro-inflammatory cytokines, which can sustain inflammation in many organs, including the gut [1,3]. Inflammation is the natural immune response to tissue damage in the human body [11]. It is initially protective for the removal of the injurious stimuli and damaged tissues, as well as the initiation of tissue healing [11]. However, it becomes problematic when there is overnutrition, which makes inflammation chronic. This is because the nutrition from the degradation of the damaged tissue, together with the excessive nutrition, will be mostly turned into lipid intermediates and deposited in healthy non-adipose tissues, causing lipotoxicity and further tissue damage [12,13]. Thus, over-nutrition will lead to a vicious cycle of excessive lean mass (such as protein) broken down and lipid intermediates piling up, fueling chronic inflammation. For its positive influence on gut microbiota and the immune system, the MD is now proposed not only as a potential tool in the management of different medical conditions, but also for health promotion and prevention globally [14]. Epidemiological evidence, in fact, has demonstrated that adherence to the MD can greatly reduce the risk of overall mortality and the incidence of many conditions such as cancer, diabetes, respiratory and cardiovascular diseases [15,16], inflammatory and functional diseases and obesity [17,18]. The aim of this narrative review was to determine the role of the MD in several gastrointestinal diseases, so as to give a useful overview of this relationship, identifying a new nutritional approach for these disorders.

We performed a literature review using the PubMed database in order to find articles regarding the MD as a potential tool in prevention and management of gastrointestinal diseases. We carried out three different researches regarding the association between MD and inflammatory bowel diseases, MD and functional gastrointestinal diseases and MD and coeliac disease. We searched the articles using key-terms commonly related to IBD (Crohn’s disease, ulcerative colitis, adalimumab, infliximab, abdominal pain, etc.), FGIDs (irritable bowel syndrome, functional dyspepsia, functional constipation, functional abdominal pain, Rome IV criteria, etc.), and coeliac disease (gluten-free diet, immune-based enteropathy, etc.), combined with terms to describe the MD (antioxidants, vitamins, etc.). We reviewed a variety of different types of studies, including, for example, randomised controlled trials, systematic reviews, interventional studies, and observational studies. The data found were principally related to the adult population, although paediatric studies were also included.

## 2. The Mediterranean Diet and Inflammatory Bowel Diseases

Inflammatory bowel diseases (IBD) are chronic autoimmune disorders characterised by chronically relapsing–remitting inflammation of the gastrointestinal (GI) tract [19]. The prevalence of IBD exceeds the 0.3% in Oceania, North America and in many European countries, and the incidence of these diseases has been substantially increasing in the last decades, principally in industrialised countries, making IBD an important burden on health systems [3,19].

The etiology of IBD is still unknown; in fact, it is a multifactorial disease [19]. Recent evidence indicates that a combination of susceptible genes, environment, altered microbial flora and inappropriate immune responses might be factors involved and functionally integrated in the pathogenesis of IBD [19]. As IBD incidence and prevalence both in the adult and in the paediatric population are constantly increasing [20], new data emerging from developing countries and migration studies suggest that environmental factors might play a major role [21]. Therefore the identification of IBD environmental risk factors and new therapies remain subjects of intensive research, and diet is one of the best candidates [3,19]. The intestinal microenvironment is indeed normally influenced by many factors in which diet plays a vital role, as it impacts its function, the gut epithelial-mucosal layer, the microbial composition, and the immune homeostasis by direct and indirect mechanisms [8]. It is known that diet can play a role in the generation of inflammation, by modulating the microbiome, the function of the intestinal barrier and the mucosal layer [22]. Several epidemiological studies discuss a positive correlation between the “Western diet”, with high amounts of unsaturated fatty acids, proteins, high sugar loads and a low vegetable and fruit intake, and the risk of developing IBD [22,23]. The components of the Western diet such as animal fats, sugars, wheat proteins, emulsifiers and maltodextrins, can determine an altered bacterial clearance at the intestinal level, promoting bacterial adhesion and penetration and subsequent intestinal inflammation [23]. Roberts et al. recently highlighted a clear correlation between the annual emulsifier consumption (in food and beverages) and the incidence of IBD [24].

A paediatric study recently confirmed a profound imbalance in fat, vegetables, and fruits consumption and the development of CD, favouring this hypothesis [25]. In IBD the nutritional approach represents a valid therapeutic option by regulating the inflammatory mechanisms [26]. Exclusive enteral nutrition (EEN) is the first-line treatment for induction of remission in patients with Crohn’s Disease [26]. The latest ESPGHAN guidelines show that the patient with IBD must not be subject to particular food restrictions and that an individual diet must be assessed on the basis of the nutritional imbalances of the individual patient, in order to avoid aggravating the symptoms [26]. Several other diets have been suggested for management of IBD, including the low-FODMAPs diet, a gluten-free diet, and a vegan diet [26]. Nevertheless, none of these diets should be recommended to children and adolescents with IBD at present [26]. On the other hand, in recent years, much attention has been given to the role of MD in IBD [19,21,26].

Multiple research studies agree on its preventive and therapeutic role [9,27,28,29,30,31,32]. Initial interest in this diet scheme arose from the observation of lower rates of chronic and degenerative diseases, such as IBD, in the Mediterranean region compared to Western countries [9]. There are several proposed mechanisms of action to explain the association between IBD and MD. These proposed mechanisms involve the direct effect of dietary antigens, the alteration of gut permeability, and the autoinflammatory response of the mucosa due to changes in the microbiota [27]. Among the literature reviewed, we found several papers analysing the connection between single dietary elements and IBD development; to summarise, many studies agreed that a high intake of animal protein, linoleic acid and sweets and a low intake of fruits and vegetables were all risk factors for IBD development [28,29,30]. However, only a few of the studies we analysed focused their attention on the role of specific dietary patterns, such as MD in IBD prevention [28,29,30]. In 2016, the Epic Study was the first major European case-control study to analyse the link between adherence to the Mediterranean diet and the risk of developing UC or CD [31]. Between 1992 and 2000, the EPIC-IBD study gathered a group of 366,351 healthy people (male and female, aged 20–80 years) from seven European countries [31]. During the follow-up, which lasted from 2004 to 2010, incident cases of IBD were tracked down, and for each case, four controls were chosen [31]. For all the people enrolled, food-frequency questionnaires were administered at baseline, and an adapted Mediterranean-diet score (aMED) was calculated, to establish MD adherence [31]. Three distinct dietary patterns were separately generated for UC and CD [31]. The study showed that a diet rich in sugar and soft drinks was directly associated with UC risk when the disease was diagnosed after at least two years of diet recording; nevertheless, this bond failed if the vegetables intake was high, as if they offset each other [31]. On the other hand, the study was not able to prove any connection between diet and CD risk [31]. Subsequently, Khalili et al. gathered a group of 83,147 Swedish middle-aged male and females born between 1914 and 1952 and calculated for each of them a modified Mediterranean-diet score (mMED); incident cases of UC and CD were verified from the Swedish Patient Register [9]. Statistical analysis of this prospective cohort study showed that a higher mMED score was linked to a lower risk of developing CD (*p* = 0.02). Moreover, the relationship between diet and CD was confirmed, as people with poor adherence to the MD had an adjusted population-attributable-risk of 12% for later onset CD [9]. However, contrary to the work of Racine et al., no association was demonstrated between the mMED score and UC [9,31]. In addition to disease prevention, the MD seems to play a pivotal role in the management of the disease itself [32]. Its anti-inflammatory and antioxidant properties and its capacity to downregulate inflammatory pathways led researchers to investigate its effect on plasmatic and faecal inflammatory markers, such as CRP, FC and interleukins [32]. Several studies confirmed an anti-inflammatory role for MD in the healthy population, as observed in the study by Sureda et al., which concluded that a high adherence to the MD pattern was directly associated with a better profile of plasmatic inflammatory-markers (higher levels of adiponectin and lower levels of hs-CRP, TNF-α, leptin and PAI-1), particularly in healthy male adults [33]. In a six-week pilot study, Marlow et al. highlighted a small reduction of established biomarkers, such as CRP (statistically non-significant) and an altered expression of 3551 genes analysed by transcriptomics [34]. Other interesting data that emerged from this manuscript was the fact that MD could influence the microbiota composition, leading to an increase in *Bacteroidetes* and *Clostridium* clusters and to a decrease in *Proteobacteria* and *Bacillaceae*, as was seen in the normal microbiome of non-IBD patients [34]. Regarding this last topic, Illescas et al. also confirmed the protective role of the microbiota associated with the MD [35]. They concluded that following an MD lead to an increase in beneficial anti-inflammatory bacterial species, in contrast to what happens in IBD patients [35]. Moreover, Chicco et al. evaluated the impact of a short-term dietary intervention based on the adoption of the MD in 142 adult patients with a diagnosis of IBD for at least 6 months and in active follow-up [36]. After six months of a hypocaloric MD properly prescribed by a nutritionist, they observed a statistically significant reduction in BMI (UC patients *p* = 0.002, CD patients *p* = 0.023), waist circumference (CD patients *p* = 0.04) and a substantial improvement of liver steatosis (UC patients *p* = 0.016 and CD patients *p* < 0.001). Interestingly they also noticed a reduction of inflammatory markers such as CRP (UC patients *p* = 0.013, CD patients *p* = 0.035) and FC (UC patients *p* = 0.049, CD patients *p* = 0.035) [36]. This prospective interventional study also highlighted an important increase in quality of life (*p* < 0.001), which is often compromised in IBD patients. Accordingly, an MD has also been associated with an improvement in nutritional status and with a reduction of pro-inflammatory visceral fat, when associated with energy restriction [36]. This feature can be very interesting, considering that IBD patients have been shown to have an increased risk of developing non-alcoholic fatty liver disease (NAFLD) and metabolic syndrome. The MD could also have a therapeutic role in the post-surgical phase of UC patients who undergo pouch surgery. In fact, as shown in the study of Godny et al., high adherence to an MD was directly associated with lower faecal-calprotectin levels and, in the long run, lower risk of developing pouchitis [37].

Other studies focused instead on the self-perceptive quality of life of IBD patients and its correlation with a healthy and balanced lifestyle which includes adherence to the MD [5,10]. For example, Ruano et al. reported a firm association between MD adherence and some aspects of a self-perceived mental and physical quality of life [38]. The latter can also be explained by the fact that a reduction in BMI, waist circumference and a controlled healthy lifestyle might influence the perception of quality of life of these patients. A recent study by Papada et al. tried to analyse the relationship between adherence to an MD and the quality of life of IBD patients. In a cross-sectional study, they used the MedDiet score to assess patients’ dietary habits and highlighted a statistically significant correlation between adherence to an MD and improved quality of life; collaterally they also noticed an improvement in intestinal symptoms correlated with IBD (*p* = 0.008) [5]. Following a healthy lifestyle such as the MD might lead to an improvement of life quality, but can also determine a mortality reduction in patients affected by CD and UC, as seen in the work of Lo et al. [10]. Similarly, Vrdoljak et al. investigated the adherence to an MD in 94 patients aged between 18 and 65 years with an IBD-diagnosis for at least 1 year. They observed that only nine participants fulfilled the criteria for MD adherence, and all of them were male (*p* = 0.0021) [19]. On the other hand, most of the population (90.4%) considered that proper nutrition might play an important role in their health and agreed that a more controlled and better diet could reduce their IBD symptoms [19]. Indeed, most of the participants expressed their will to expand their nutritional knowledge if proper educational programs were proposed [19]. Following this example, Taylor et al., in their single-centre cross-sectional analysis, suggested the use of an MD-adherence questionnaires, such as the thirteen-item PREDIMED Mediterranean diet score to identify pro-inflammatory dietary schemes, in order to modify the patient’s dietary scheme and ameliorate their dietary intake [39]. Although many studies have been published for the adult population, little is known about the paediatric population. D’Souza et al. analysed the preventive role of diet in the paediatric population in their case-control study, trying to determine the connection between certain dietary patterns and risk of CD in Canadian children [40]. In this published paediatric case-control study, a positive association with CD was found in girls with a diet rich in meats, fatty foods, and desserts, whereas a high intake of vegetables, fruits, olive oil, fish, grains, and nuts was inversely associated with CD in both sexes [40]. Our published manuscript focused on the role of environmental factors in IBD and showed that the Mediterranean dietary pattern may exert a protective role in the development of IBD [41]. We found that low adherence to an MD was higher for CD and for UC when compared with controls [41]. In addition, El Amrousy et al. proved that the reduction of inflammatory biomarkers in the paediatric population was similar to the adult one if adherence to MD was high [42]. They analysed the positive effects of an MD in IBD patients with active CD and UC; after a 12-week diet, the clinical remission rate, as well as most inflammatory markers (CRP, calprotectin, TNF- α, IL17, IL 12 and IL13) were significantly improved, in contrast to the patients who followed their regular diet [42]. We investigated the relationship between inflammation and dietary behaviours in IBD children, in particular adherence to an MD, in comparison with a healthy control group [43]. We observed that there was a different kilocalorie intake between IBD patients and the control group (*p* = 0.024), and comparing UC to CD, there emerged a significant difference in protein intake (*p* = 0.047), iron intake (*p* = 0.023) and vitamin D (*p* = 0.044), which was higher in CD patients. Interestingly, we found a significant association between adherence to an MD and a lower level of FC in IBD patients (*p* = 0.027) [43].

Table 1 summarizes the main studies regarding the impact of the MD on IBD.

## 3. The Mediterranean Diet and Functional Gastrointestinal Disorders

Functional gastrointestinal disorders (FGIDs) affect up to 40% of the population globally at some time in their life and represent almost 50% of paediatric gastroenterologists’ consultations [44,45]. The pathogenesis is multifactorial and still unclear, as no structural or biochemical aberrations have been found [45,46,47,48]. To the present date, the biopsychosocial model could explain the correlation between several factors involved in the FGIDs aetiopathogenesis, such as genetic predisposition, altered gut–brain axis, altered gut motility, gut hypersensitivity, intestinal inflammation/infection, altered microbiome composition, psychological conditions and environmental triggers, such as food [46,49,50]. As FGIDs are extremely common among adolescents and children, they are frequently associated with functional disability, impaired quality of life, anxiety, school absenteeism and a remarkable increase in health-care costs [46]. Currently, paediatric and adult functional gastrointestinal disorders are described as separate but overlapping diseases within the Rome IV criteria [51]. Irritable bowel disease (IBS) represents 65% of the diagnosis, followed by FAP, abdominal migraine, functional dyspepsia, functional constipation, and diarrhoea [52]. Our literature analysis reviewed the most frequent FGIDs diagnosed in clinical practice, such as IBS, functional abdominal pain (FAP), functional dyspepsia (FD) and functional constipation (FC) and their association with MD. FGIDs management remains principally symptom-based, since the exact pathophysiological mechanisms are not completely understood [46]. Various treatment options have been proposed, such as fibre supplementation, probiotics, cognitive behavioural therapy, psychosocial interventions, antidepressants, antispasmodics and prokinetics [53]. There are indications that psychological factors, including anxiety and stress, may increase the severity of symptoms related to FGIDs through the gut–brain axis [53,54]. Regarding IBS treatment, studies show that a combination of medical treatment and psychotherapy, especially cognitive behavioural therapy, has a significant response in decreasing the symptoms [55]. However, diet is considered a cornerstone in FGID treatment, in particular for IBS, as diet-based therapies have shown efficacy in improving symptoms and may lead to a better quality of life and better clinical health outcomes [56]. In fact, nutrients can play a pivotal role in gut microbiota, modifying the prevalence of some phyla over others, and modulating the function of the intestinal barrier and intestinal motility. Food is a key element in the patient’s life, as it can act both as a trigger and as a cure. In fact, it has been demonstrated that the majority of children (up to 93%) with FGIDs identify certain food to be the cause of a symptom exacerbation such as diarrhoea or abdominal pain, and, conversely, certain foods ameliorate their symptoms [44,56]. Recently, more interest has been focused on low fermentable oligosaccharides, disaccharides, monosaccharides, and a polyol (FODMAP) diet, in which the intake of these fermentable carbohydrates is reduced [57].. In contrast, emerging evidence supports the hypothesis that the MD, which is rich in FODMAPs, may be beneficial for FGIDs [48]. As seen in many studies, adherence to the MD, in fact, can determine a remarkable reduction in all-cause mortality, especially in cardiovascular diseases and cancer, but in inflammatory and functional ones as well. As regards functional abdominal pain and functional constipation, in the literature reviewed there were no high-quality studies which could identify an effective dietary scheme, especially for children [58]. One of the reasons might be related to the multifactorial pathogenesis of these diseases, which make it difficult to individuate an efficacious dietary pattern [52,59]. Nevertheless, these patients still frequently seek information about how they can change their diet scheme in order to improve their quality of life and decrease their symptoms [58,59]. Taking this limit into account, we analysed the relationship between certain foods that are staples of the MD and their role in FGIDs prevention and development (FAP and FC in particular). The frequent assumption of a fresh-vegetable, fruit and high-fibre intake, is one of the cornerstone features of the MD [60]. Vegetables, in fact, are relevant components of MD, rich in phytosterols and flavonoids, while on the other hand, fruits are a precious source of terpenes, minerals, and antioxidant factors, as well as flavonoids. Furthermore, this dietary pattern often contemplates foods rich in polyunsaturated fatty acids (PUFA), nitrate/nitrite/NO and polyphenolic agents, which have a vasodilatory effect, anti-inflammatory, immunomodulating, antioxidant properties and improve tissue micro-regulation. [48,61,62]. The MD is also effective in increasing gut-beneficial bacterial species, thus creating an anti-inflammatory environment [62]. It can also be assumed that its benefits do not derive from single nutrients contained in this diet scheme, but from their synergistic actions [63].

As demonstrated by Oyebode et al., in adult patients, eating seven pieces of fresh fruit and vegetables per day greatly benefits general health and can improve functional-disorder symptoms [64]. These data were also confirmed in a 4-week comparative effectiveness trial conducted by Ansell et al., which enrolled 79 patients affected by functional constipation; the daily consumption of kiwifruit and prunes led to a significant increase in stool frequency and improvement of gastrointestinal symptoms [65]. Confirming the above considerations in the paediatric population, Malaty et al. showed that the prevalence of FAP was higher in children who ate small quantities of fruits and vegetables every week [66]. Accordingly, Malaty et al., in a community-based cross-sectional study of 925 children, highlighted an inverse correlation between fruit consumption and FAP prevalence, showing a higher incidence of FAP in patients with a lower fruit-intake, even if this association was not demonstrated in the same study as regards the consumption of fresh vegetables [66]. Insufficient dietary-fibre intake has been strongly associated with FGIDs in several studies, even though studies related to fibre supplementation in children with functional disorders have reported mixed results [44]. Fibre, in fact, with its water-retention properties, can accelerate colonic-transit time and can play a pivotal role in normalising stool form and modifying gut microbiota [53]. A lower fibre-intake in children with FC has been widely demonstrated in numerous articles. Moreover, in patients with refractory constipation, attempting to introduce and maintain a high-fibre dietary pattern is extremely hard [67]. Miranda et al. asserted that the consumption of fibres below the daily recommended amount can be stated as a risk factor for developing FAP [59]. In addition, Feldman et al. concluded that a 10 g corn-fibre supplementation for two weeks was much more effective than a placebo in reducing gastrointestinal discomfort in FAP patients [68]. In line with this, and focusing instead on the paediatric population, several studies reported that fibre supplementation in children with functional abdominal diseases was statistically significantly more effective than a placebo in reducing abdominal-pain frequency [56]. These results were not confirmed in all the studies reviewed. Horvath et al. in his two double-blind randomised controlled trials, which aimed to find a beneficial association between fibre intake and the reduction of FAP and IBS symptoms, did not find any difference between the treated and the placebo group [69]. The MD is characterised by the consumption of small but frequent meals, particularly beneficial for functional dyspepsia. Its low content of saturated fats may ease gastric emptying and promote a limited stimulation of the gastro-colonic reflex, which is often overstimulated in FGIDs [48,60]. In addition, a high-fat meal is proved to be associated with an effective decrease in the gastric-emptying rates, as demonstrated by Parker et al. in his prospective cross-sectional study, where a low-fat and low-calorie dinner induced significantly fewer dyspeptic symptoms, compared with a high-fat meal [70]. In a randomised study, Feinle-Bisset et al. also confirmed that high-fat foods elicited more dyspeptic symptoms, compared with low-fat foods [71]. Regarding the MD as a dietary pattern, Zito et al. aimed to see if there was a link between adherence to the MD and onset of FGIDs, such as IBS and FD. They gathered a population of 719 healthy controls, 172 people with IBS and 243 people with FD, in accordance with the Rome III criteria. And all these patients were further divided into age groups: 17–24 years old, 25–34 years old, 35–49 years old, 50–64 years old and over 64 years old [48]. Adherence to the MD was calculated using different food-frequency questionnaires, in accordance with the patient’s age. Their multivariate analyses showed that low (*p* < 0.0001) and intermediate (*p* < 0.05) MD-adherence was associated with IBS and FD (in this case, only in the youngest group). On the other hand, in the older group MD-adherence was high, despite the persistence of functional gastrointestinal symptoms. The authors explained this data by referring to the fact that gastrointestinal symptoms might be due to other factors, such as ageing, medication and comorbidities [48]. Similarly, Elmaliklis et al. concluded that high adherence to the MD could be a protective factor in gastro-intestinal-disease development [63]. In their retrospective, randomised case-controlled study, they gathered a group of 142 patients with different gastrointestinal diseases (UC, CD, IBS and gastroesophageal reflux disorder) and 147 healthy controls [63]. They analysed people’s food habits, such as functional food consumption and MD adherence in the 2–3 years before diagnosis/recruitment. They saw that the MD index was higher in healthy controls compared to patients with gastrointestinal disorders, and, simultaneously, that a high intake of functional food such as probiotics, prebiotics, herbs and plant foods in the 2–3 years before diagnosis could be considered a protective factor as well [63].

MD has an important role, not only in preventing the diseases from developing, but also as a treatment option. Since the low-FODMAP diet, the gluten-free, and the balanced diet are the ones that are more often prescribed in clinical practice, Paduano et al. [72] wanted to compare their efficacy and tolerability in patients diagnosed with IBS. The authors concluded that all three were effective in ameliorating IBS severity and abdominal bloating and in lessening the duration of the abdominal pain. However, the low-FODMAP diet was the only one that proved to be effective in normalising stool solidity [73]. The low-FODMAP diet and gluten-free diet (GFD) led to an improvement of the physical and mental state, while the balanced diet or the MD resulted in psychological improvement. Nevertheless, the MD proved to be the one with the highest adherence-index [72]. Altomare et al., on the other hand, concluded that IBS patients who had lower MD-adherence experienced more severe symptoms, such as abdominal pain and flatulence [73]. Collaterally, they observed that an inadequate diet, lower in reference-intake ranges for macronutrients, led to microbiota alterations [73]. Confirming what was already known in the adult population, Agakidis et al. found a protective role for the MD in FGIDs development in the paediatric population [67]. This study was conducted with a population of 1116 children and adolescents, and it showed how FGIDs were inversely associated with MD-adherence and positively associated with age [67]. On the other hand, in our cross-sectional study including a large number of children and adolescents from six countries in the Mediterranean area, we found that the prevalence of FGIDs was not related to FODMAP intake and that it varied significantly, depending on adherence to the MD. The MD could indeed be considered as a protective factor, as we found a statistically significant association between FGIDs and the KIDMED score [74]. We did not find many studies looking at the therapeutic role of MD in paediatric patients. The most promising was the one by Al-Biltagi et al., who aimed to assess the tolerance, security and efficacy of the MD in children and adolescents with IBS. In their 100-people study, half of the patients received a six-month MD with good adherence, while the other half continued their regular diet [61]. The MD was well-tolerated by the patients, and was free from adverse reactions, as well as proving effective in reducing all IBS scores (IBS-symptoms-severity-score questionnaire, IBS-quality-of-life questionnaire) [61]. Table 2 shows the main studies regarding the impact of the MD on IBS.

## 4. The Mediterranean Diet and Coeliac Disease

Coeliac disease is a multifactorial, systemic and immune-based enteropathy, characterised by a specific serum antibody-response consequent to gluten ingestion [75,76]. Gluten is a protein complex rich in folates, vitamin B and iron, which is contained in wheat, barley, rye, kamut, etc. [77]. Coeliac disease occurs in around 1–1.4% of individuals worldwide, although a higher prevalence of the disease is observed in developed countries [75,78]. Gluten ingestion can lead to damaging effects on the mucosa of the small intestine, such as villous atrophy and lymphocyte infiltrate [75,76,77]. As a result of this, genetically susceptible people manifest a various range of clinical symptoms after gluten intake, which more frequently include diarrhoea, abdominal pain and extra-intestinal manifestations, such as anaemia, fatigue, weight loss, and dermatitis herpetiformis [75,76,78,79]. Diagnosis is based on blood tests, eventually integrated into endoscopic findings and genetic analysis [78]. The only effective and available treatment is adherence to a gluten-free diet, limiting the daily gluten intake to 10–50 mg/day [80]. Although the GFD is highly effective and it is, so far, the only therapeutic option for coeliac patients, the literature underlines how nutritionally unbalanced it is, especially compared with the MD. Only few papers examined the therapeutic use of the MD in coeliac patients. The gluten-free diet principally consists of packaged foods with a high glycaemic-index, as well as a high content of carbohydrate, lipid and salt [77,79,80]. Considering that gluten-containing products are rich in folates and vitamins, it is easy to assume that many coeliac patients have mineral and vitamins deficits [77,79,80]. Other interesting data show that the GFD is poor in fibre, because during the process of food manipulation the outer layer of grain is removed [77,79,80].

The reason for such poor nutritional properties in the GFD can be related primarily to the low quality of packaged GF foods, in which manufacturers include more salt and lipids in their formulation in order to improve the palatability, making it inadequate in terms of nutritional properties and less nutritious than its gluten-containing counterparts. In addition, a high consumption of foods rich in fat and extremely caloric, in order to compensate for a restricted dietary scheme such as GFD, has also been widely reported in the literature, both in adults and children with coeliac disease [80]. For this reason, there is a rising concern about the fact that GFD may contribute to cardio-metabolic risk and to the modification of gut microbiota, predisposing patients to develop other inflammatory or functional diseases [80].

These uncertainties regarding the nutritional status of the GFD were stoked by Morreale et al. in one of the first studies assessing the adherence to the MD in coeliac patients. In their cross-sectional study, they recruited 224 participants, including 122 patients affected by coeliac disease; they observed that non-coeliac patients had a higher consumption of fruits, and, on the other hand, coeliac-disease patients had a higher consumption of potatoes, bread and processed meat [17].

Likewise, Larretxi et al. analysed the adherence to the MD of 83 children and adolescents affected by coeliac disease. They concluded that only 47% of the boys and 25% of the girls followed a MD [81]. Moreover, as in the studies on the adult population, they concluded that coeliac patients who followed a GFD had micronutrient deficiencies, as well as an unbalanced intake of macronutrients, such as fat and carbohydrate [81].

Similarly, Lionetti et al., in their large control study comparing coeliac-disease paediatric patients’ daily food intake with that of healthy individuals, described a much higher intake of fat in the coeliac-disease group, and, furthermore, significantly lower fibre consumption [82]. Another weak point of the GFD was indicated in a recent study by Nestares et al., who underlined the risk of a reduction in bone mineral density, and consequently the development of bone alterations in paediatric coeliac-disease patients [75]. The malabsorption typical of the disease and the persistent inflammatory state associated with the low mineral and vitamin intake characteristic of GFD, can determine an important reduction in bone mineral composition, predisposing the patient to future complications [75]. They conducted a prospective cross-sectional study between a total of 40 non-coeliac children compared with 59 children diagnosed with coeliac disease. The lean bone and fat mass of the study population were measured through dual-energy X-ray. They observed that adherence to MD, along with effective physical activity, were associated with a higher Z-score and lean mass in instrumental analysis; therefore, they proposed MD-adherence as a potential protective factor in preventing bone damage [75]. As a possible solution to the inadequate nutritional status of GFD, Bascunan et al., proposed a combination of the MD pyramid developed by the Italian Paediatric Society, rich in fibre, antioxidants and low in fat, and the GFD, as a promising alternative to fill the gap and reach a healthy dietary pattern in the context of a restrictive and gluten-free diet [80]. Table 3 reports the main studies regarding the use of MD in coeliac disease management.

## 5. Future Perspective

The MD has been increasingly proposed as a health-protective diet because of its efficacy in the reduction of all-cause mortality and morbidity, especially in cardiovascular, neoplastic and inflammatory diseases. As regards IBD, different outcomes may also rely on the patient’s variability, including genetic susceptibility and the influence of environmental factors and the disease progression itself. First of all, the age of the patients may influence intestinal microbiota and health status, as well as different genotypes correlating with better or worse course and prognosis. Moreover, changes in pharmacological therapies that might have occurred during the observational period might have overestimated the effects of diet on IBD symptoms. However, considering the potential beneficial effects of the MD together with the safety of its usage, even in the paediatric population, the potential impact of this intervention could be significant. From the opposite point of view, the relationship between food and FGIDs is well known, as many of the patients complain that certain foods act as trigger to the symptoms; it is therefore natural to think that certain dietary schemes can be used to treat the disease itself. Recommendations such as decreasing fat intake, avoiding alcohol and coffee intake and adopting a slower and more regular eating-pattern must be recommended to these patients as a potential tool in managing their symptoms. In addition, even if the low-FODMAP diet has been a cornerstone therapeutic option up until now, the MD represents a possible valid and more economical alternative, even though the studies regarding its application are lacking, especially the ones that involve a large population. Further studies are necessary in order to propose a demonstrated and applicable dietary scheme which can be used in daily clinical practice. As regards patients affected by coeliac disease, it is an absolute truth that the one and only option for treating the disease is represented by following a strict gluten-free diet. However, it is important to understand and know its weak points, such as the nutritional inadequacy of the GFD, which emerged in the studies we reviewed. Characterised by its high intake of fat and sugar associated with a low-intake of fibres, the gluten-free diet represents an unbalanced dietary pattern, which can expose patients to potential long-term outcomes. For this reason, it is important to highlight the need for coeliac disease patients to receive a personalised nutritional-education programme and to be introduced to a more structured and healthier dietary pattern, such as the MD. One of the goals could be reducing the use of processed GF food through an increase in the consumption of natural GF products which are present in the MD, thus ameliorating the nutrient intake during childhood and embracing a healthier dietary pattern.

## 6. Conclusions

As seen in these recent evidence, we believe that MD is stepping up as one of the most encouraging dietary choices for gastrointestinal diseases, and perhaps should be brought more closely to patients via structured nutritional-educational programs. A future approach could be represented by the development of a patient-shaped dietary pattern which could meet the patient’s needs and provide a balanced, anti-inflammatory and healthy diet, associated with the standard pharmacological treatments. Even if results from studies are encouraging, further studies are necessary. Some possible biases of the available literature are related to the fact that most of the papers focused on the administration of food-frequency questionnaires that the patients filled in retrospectively. Another possible bias is connected to those studies which did not have a healthy control group to compare with. In addition, there is no sufficient evidence regarding the prolonged use of the MD in patients affected by gastrointestinal disorders. However, the MD remains a promising approach, whose use needs to be evaluated in large-scale studies with a longer observation time, in order to evaluate its effectiveness in prevention and management.

## Figures and Tables

**Table 1 nutrients-15-00079-t001:** Main characteristics of the studies regarding the impact of a Mediterranean diet on Inflammatory Bowel Disease.

Study	Type of Study	Population	Aim	Results
Marlow et al. [34], 2013	Prospective study	Eight adult patients with active stable Crohn’s disease (six females and two males, aged between 31 and 60 years).	Evaluation of changes in inflammation and in the gut microbiota after administration of a 6-week Mediterranean-inspired diet.	Small reduction of inflammatory biomarkers (non-statistically significant). A total of 3551 genes had significantly (*p* < 0.05) altered expression as a result of the dietary intervention. Normalising trend of the microbial gut composition.
Racine et al. [31], 2016	Prospective study	A total of 366,351 adult patients (20–80 years). During the follow-up, 117 patients developed CD, while 256 developed UC.	Evaluate the connection between adherence to the MD, assessed by an adapted Mediterranean diet score (aMED), and the risk of developing UC and CD.	A diet rich in sugar and soft drinks was positively associated with UC, when diagnosed at least 2 years after diet recording. No association between any dietary pattern and CD risk.
Sureda et al. [33], 2018	Observational study	A total of 598 healthy patients (364 adolescents and 234 adults).	Evaluate the connection between inflammatory biomarkers and MD adherence (using food-frequency questionnaire (FFQ) and two 24 h diet recalls).	High adherence to the MD was associated with a better inflammatory-biomarker profile.
Godny et al. [37], 2019	Prospective observational study	A total of 153 adult patients with UC, who underwent pouch surgery.	Evaluate the connection between adherence to the MD, using FFQ and inflammatory markers (CRP and faecal calprotectin) and pouchitis-disease-activity index (PDAI).	MD-adherence was higher in patients with inactive disease. High MD-adherence was inversely associated with elevated calprotectin and lower risk of developing pouchitis in the years after surgery.
Papada et al. [5], 2019	Cross-sectional study	A total of 86 CD adult patients (45 in remission and 41 in relapse).	Evaluate the adherence to MD in patients with CD and assess the role of MD in improving intestinal symptoms and inflammatory markers.	Adherence to MD was greater in patients with inactive disease. The MedDiet score correlated positively with the inflammatory bowel disease questionnaire (IBDQ), and negatively with disease activity.
Khalili et al. [9], 2020	Prospective study	A total of 83,147 patients.	Evaluate the connection between MD adherence, using semiquantitative food-frequency questionnaire (SFFQ) and IBD risk.	A higher mMED score was linked to a lower risk of developing CD. People with poor adherence to an MD had an adjusted population-attributable risk of 12% for later onset CD. No association was proved between UC and mMED.
Vrdoljak et al. [19], 2020	Cross-sectional study	A total of 94 adult patients (44 in the UC group and 50 in the CD group) with IBD diagnosed for at least 1 year.	Investigate nutritional habits and dietary attitudes in IBD patients, in addition to assessing their adherence to the MD.	Only nine participants fulfilled criteria for MD adherence, all of them male. A total of 86.2% of subjects considered certain foods as responsible for exacerbating their gastrointestinal symptoms. A strict correlation between Mediterranean diet serving score (MDSS) and HDL cholesterol levels was observed. Most of the population (90.4%) considered that proper nutrition plays an important role in their illness and quality of life, and considered that a more controlled and better diet could reduce their IBD symptoms.
Chicco et al. [36], 2021	Prospective, Interventional study	A total of 142 patients (18 years old and older) with diagnosis of IBD for at least 6 months in active follow-up.	Impact of short-term dietary intervention based on the adoption of MD on anthropometric parameters, serum lipid profile, liver function and intestinal disease activity.	Improvement of anthropometric measures (BMI and waist circumference reduction), decrease in fat body mass and increase in lean body mass (no statistical significance). Significant improvement of liver steatosis and liver function (reduction of alanine aminotransferase and gamma-GT within the reference range). Improvement of inflammatory biomarkers and of quality of life.
Taylor et al. [39], 2018	Single-centre cross-sectional study	A total of 67 patients affected by inactive CD (mean age of 45).	Investigate micro- and macronutrient intake and dietary attitudes in CD patients, including their adherence to the MD, and compare them to a representative of healthy individuals.	Patients with CD had multiple vitamin and micronutrients deficits and lower MD adherence, compared with healthy controls.
D’Souza et al. [40], 2008	Case-control study	A total of 149 cases of children (2.6–20 years) with CD, 251 controls.	Obtain gender-specific dietary patterns and calculate their associated risk for CD.	Dietary pattern rich in vegetables, fruits, dairy products, eggs, olive oil, dark breads, grains, fish and nuts was negatively associated with CD, for both boys and girls.
Strisciuglio et al. [41], 2017	Case-control study	A total of 264 patients (1–18 years); 102 children with CD and 162 children with UC, and 203 healthy controls.	Evaluate the role of environmental factors in IBD development. *	Low adherence to MD was higher for CD and for UC when compared with controls.
Strisciuglio et al. [43], 2020	Single-centre cross-sectional study	A total of 125 children with a diagnosis of IBD in clinical remission and 125 healthy controls.	Assess dietary intake through a 3-day food diary and Mediterranean-diet-quality index for children and adolescents (KIDMED); evaluate their adherence to MD and investigate the relationship between inflammation and dietary behaviours.	IBD patients and healthy controls had an intermediate adherence to MD. IBD group had a higher kilocalorie intake. Significant association between adherence to MD and a lower level of FC in IBD patients. In comparing CD and UC, a higher intake of protein, iron and vitamin D in CD patients emerged.
El Amrousy et al. [42], 2022	Prospective randomised study	A total of 100 patients (12–18 years) with mild-moderately active CD or UC. Group I (26 patients with active CD and 24 patients with active UC) received Mediterranean Diet for 12 weeks. Group II (28 patients with active CD and 22 patients with active UC) followed its regular diet.	Evaluation of clinical remission, clinical scores (PCDAI and PUCAI) and inflammatory biomarkers (CRP, calprotectin, TNF-alfa, IL17, IL12, IL13).	Most of the patients reached clinical remission after a 12-week diet. Clinical scores (PCDAI and PUCAI) and inflammatory markers were significantly improved in patients in Group I (all cytokines were decreased, except IL10).

* Among the various areas examined in the study, we focused our attention on dietary habits.

**Table 2 nutrients-15-00079-t002:** Main characteristics of the studies regarding the impact of Mediterranean diet on FGIDs.

Study	Type of Study	Population	Aim	Results
Zito et al. [48], 2016	Case-control study	A total of 1134 people (598 males and 536 females, 17–83 years).	Assess the connection between MD adherence and onset of functional gastrointestinal disorders (IBS and functional dyspepsia).	MD adherence is inversely associated with the prevalence of gastrointestinal symptoms.
Elmaliklis et al. [63], 2019	Retrospective case-control study	A total of 289 adults (127 males and 162 females); 142 patients had GI disorders (UC, CD, IBS or gastroesophageal reflux disease) and 147 healthy controls.	Investigate the connection between functional food consumption and MD adherence in gastrointestinal-disorders development.	The MD index was higher in controls. In the 2–3 years before diagnosis, patients consumed less functional food (for example, probiotics, prebiotics, vegetables), than controls.
Paduano et al. [72], 2019	Prospective interventional study	A total of 42 patients (18–45 years old) diagnosed with IBS, in accordance with the Rome IV criteria.	Compare the efficacy of three diets (low-FODMAP, gluten-free diet and balanced diet) in IBS treatment.	The low-FODMAP diet proved to be effective in normalising stool solidity. All three diets were effective in reducing disease severity, abdominal bloating and the duration of abdominal pain. The low-FODMAP and the gluten-free diet improved the quality of life of the patients both physically and mentally, while the balanced diet led only to a psychological improvement.
Altomare et al. [73], 2021	Cross-sectional study	A total of 28 IBS patients (median age 55 years old) and 21 controls (median age 56 years old).	Evaluate the association between MD adherence, using MDSS, and macronutrients intake with microbiota composition and clinical symptoms, in IBS patients.	IBS patients had a lower MD-adherence than controls, in addition to an inadequate diet. These patients complained about more severe symptoms (abdominal pain and flatulence). IBS patients with an inadequate diet had an altered microbiome composition.
Agakidis et al. [67], 2019	Prospective cohort study	A total of 1116 children and adolescents (6–18 years old).	Assess the link between MD adherence, using KIDMED score, and FGID development, using Rome III criteria.	FGIDs were inversely associated with MD-adherence, and positively associated with age.
Strisciuglio et al. [74], 2022	Multicentre cross-sectional study	A total of 4422 people (4–18 years), divided into: Group A: 1972 children aged 4 to 9 years; Group B: 2450 adolescents aged 10 to 18.	Investigate the connection between FGIDs prevalence in Mediterranean area and FODMAP intake and MD adherence.	No association was found between FGIDs and FODMAP intake. A statistically significant association was found between FGIDs and the KIDMED score. A statistically significant association was found between MD adherence and FC and PDS.
Al-Biltagi et al. [61], 2022	Prospective, randomised cross-sectional controlled study	A total of 100 people with IBS (12–18 years old). Group 1 (50 patients): Mediterranean diet administration with good adherence; Group 2 (50 patients): regular diet.	Assess the tolerance and security of the MD in children and adolescents with IBS. Evaluate the MD efficacy as treatment.	MD was well tolerated; no adverse reactions to, or effects of, the diet were reported. IBS symptoms severity-score questionnaire, IBS quality-of -life questionnaire were significantly improved in the group who received the MD.

**Table 3 nutrients-15-00079-t003:** Main characteristics of the studies regarding the use of the Mediterranean diet in Coeliac disease management.

Study	Type of Study	Population	Aim	Results
Morreale et al. [17], 2018	Cross-sectional study	A total of 122 adult coeliac patients who followed a GFD, and 102 healthy adults.	Evaluation of MD adherence.	Coeliac patients had a low adherence to the MD. The nutritional status of coeliac patients was poorer than the control group. Coeliac patients ate higher amounts of potatoes and red meat, while the fruit intake was higher in the control group.
Larretxi et al. [81], 2018	Prospective study	A total of 83 paediatric coeliac patients (3–18 years old).	Assessment of the nutritional adequacy of the GFD.	A total of 47% of the boys and 25% of the girls followed an MD. Coeliac patients who followed a GFD had micronutrient deficiencies, as well as an unbalanced intake of macronutrients (fat and carbohydrate).
Lionetti et al. [82], 2020	Prospective case-control study	A total of 120 coeliac children (4–16 years old) on a GFD for more than 2 years, and 100 healthy children as controls.	Assessment of the nutritional status of the patients, as well as their dietary habits and their adherence to the MD.	Coeliac patients’ nutritional status is not dissimilar from the healthy control group; however, their diet is unbalanced (rich in fats and poor in fibres). The KIDMED index is 6.5 in coeliac patients, versus 6.8 in healthy controls.
Nestares et al. [75], 2021	Prospective cross-sectional study	A total of 59 children with coeliac disease (7–18 years old), and 40 healthy controls.	Evaluation of the influence of MD-adherence and physical activity on bone health in paediatric coeliac patients.	Adherence to the MD, along with effective physical activity, were associated with a higher Z-score and lean mass in instrumental analysis.

## Data Availability

Not applicable.

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
