# Peer review of "The Mediterranean Diet in Paediatric Gastrointestinal Disorders"

_nutrients, 2022, doi:10.3390/nu15010079_

Round 1
Reviewer 1 Report
Dear authors,
This is an important topic and good summary. I have no major recommendations other than to review the manuscript carefully for proper punctuation, grammar and sentence structure. I have listed some corrections below. Tables also need to be modified. As they currently stand, they are not easy to follow or read.
- Lines: 20-22. Consider sentence revision
- Line 53, patients should be “patient”
- Line 52: long sentence. Consider dividing into two sentences
- Line 58-59, add commas after “system”, proposed, conditions.
- Line 81: should it read 0.3%?
- Line 85: disease has an 18 after it. Thinking it is likely a reference?
- Line 89: “are” instead of “is”
- Line 114: FODMAP instead of FOODMAP
- Line 118: multiple works means what? Multiple papers? If so, there should be more than one reference cited with such a bold statement.
- Line 132: 366,351 instead of 336.351
- Line 153: consider using “Several” vs “many”
- Line 162: italicize names of bacteria
- Line 182: capitalize “High”
- Line 258: aside from diet, behavioral health and congntive behavioral therapy is just as important in management. I would acknowledge this.
- Line 298: consider changing to, “but from their synergistic actions.”
- Line 316. Would place a “.” After “numerous articles”. Start a new sentence with “Moreover,…”
- All Tables need to be reviewed for spacing, careful to not cut off words. Remove bullets in table 2. Columns need to be widened to make it easier to read tables.
- Line 484: would consider avoiding “common sense” recommendations. Sounds arrogant
Reviewer 2 Report
Comments to “Mediterranean Diet in paediatric Gastrointestinal disorders”
This review paper investigated the possible protective effect of Mediterranean diet in reducing all-cause mortality and morbidity, especially in cardiovascular, neoplastic and inflammatory diseases like chronic gastrointestinal diseases. Yet, the manuscript contains fundamental misunderstanding about inflammation.
Inflammation is the natural immune response to tissue damage in the human body [1]. It is initially protective for the removal of the injurious stimuli and damaged tissues as well as the initiation of tissue healing [1]. Inflammation becomes problematic only when there is overnutrition which makes inflammation chronic. This is because, the nutrition from the degradation of the damaged tissue together with the excessive nutrition already existed inside the body will be mostly turned into lipid intermediates and be deposited in healthy non-adipose tissues, causing lipotoxicity [2,3] and further tissue damage. Thus, over-nutrition will lead to a vicious cycle of excessive lean mass (like protein) broken down and lipid intermediates piling up, fuel chronic inflammation.
Reference:
1. Costantini S, Sharma A and Colonna G (2011). The Value of the Cytokinome Profile, Inflammatory Diseases - A Modern Perspective, Dr. Amit Nagal (Ed.), ISBN: 978-953-307-444-3, InTech, Available from: http://www.intechopen.com/books/inflammatory-diseases-a-modern-perspective/the-value-of-the-cytokinome-profile
2. Garbarino, J.; Sturley, S.L. Saturated with fat: New perspectives on lipotoxicity. Curr. Opin Clin. Nutr. Metab. Care 2009, 12, 110–116.
3. Saltiel, A.R.; Olefsky, J.M. Inflammatory mechanisms linking obesity and metabolic disease. J. Clin. Investig. 2017, 127, 1–4.
Round 2
Reviewer 2 Report
As the authors have addressed all my concerns about the manuscript stated in my previous review report, the authors can do some minor revision to correct the following typos in the manuscript:
1. Page 2, line 53-54, “Indeedadipose” should be “Indeed adipose”;
2. Page 3, line 134-135, “several papersanalysing” should be “several papers analysing”;
3. Page 4, line 157, “Mmoreover” should be “Moreover”;
4. Page 5, line 232, “MDin” should be “MD in”;
